# Alternative TSSs are co-regulated in single cells in the mouse brain

Kasper Karlsson[1] ID, Peter Lönnerberg[2] & Sten Linnarsson[2,*] ID

## Abstract

Alternative transcription start sites (TSSs) have been extensively studied genome-wide for many cell types and have been shown to be important during development and to regulate transcript abundance between cell types. Likewise, single-cell gene expression has been extensively studied for many cell types. However, how single cells use TSSs has not yet been examined. In particular, it is unknown whether alternative TSSs are independently expressed, or whether they are co-activated or even mutually exclusive in single cells. Here, we use a previously published single-cell RNA-seq dataset, comprising thousands of cells, to study alternative TSS usage. We find that alternative TSS usage is a regulated process, and the correlation between two TSSs expressed in single cells of the same cell type is surprisingly high. Our findings indicate that TSSs generally are regulated by common factors rather than being independently regulated or stochastically expressed.

**Keywords** alternative TSS usage; neurons; single-cell RNA sequencing; transcription; UMI

**Subject Categories** Genome-Scale & Integrative Biology; Transcription

**Mol Syst Biol. (2017) 13: 930**

## Introduction

Our understanding of TSS usage has increased dramatically over the last decade since the introduction of deep sequencing technologies. It is now clear that most genes are transcribed from multiple distinct TSSs. For example, the FANTOM consortium recently found an average of four robust TSSs per gene, across more than 800 tissues and cell lines (Forrest *et al*, 2014); however, the number of TSS reported was highly dependent on the filtering method used and was complicated by CAGE peaks in enhancer regions, coding regions, and promoter-associated short RNA (de Klerk & t Hoen, 2015). Transcriptome-wide studies using DeepCAGE have found that the hippocampus has a larger number of active TSSs compared to other cell types tested and that generally TSS use is highly tissue specific (de Klerk & t Hoen, 2015). For example, IGF1 and IGF2 are

known to be regulated by multiple TSSs and expressed in various embryonic and adult tissues (Leroith & Roberts, 1993). In *Drosophila*, alternative TSS generally implements distinct regulatory programs during development (Batut *et al*, 2013). Alternative TSS usage can affect protein diversity by incorporating extended or alternative N-terminal polypeptides. One example is NADH-cytochrome b5 reductase, where usage of alternative TSSs results in two protein forms, one membrane-bound and one soluble form (Ayoubi, 2005). However, the majority of alternative TSS do not change the mRNA coding potential, and thus, for the majority the biological effect must come from isoform-specific regulation like mRNA abundance, stability, and localization (Rojas-Duran & Gilbert, 2012). Alternative TSS can also cause differences in translation efficiencies up to a 100-fold when examined in yeast (Rojas-Duran & Gilbert, 2012).

Transcription start sites are activated by a complex chain of events initiated by the binding of transcription factors to proximal sites or distal enhancers. However, the rules by which local or distal transcription factor (TF) binding causes the activation of specific local TSSs are not well understood. It is often assumed that TF binding upstream of a TSS will specifically activate that TSS, but it is also possible that multiple nearby TSSs could be simultaneously activated, or indeed that local and distal enhancers could exhibit preferences for specific TSSs.

Surprisingly, in general, it is not known to what extent alternative TSSs show cell type-specific expression. In the extreme case, it is possible that all TSSs respond equally well to regulatory input, that is, alternative TSSs are regulated by common factors. Alternatively, there may be hitherto unexplored rules that determine a strict preference for each regulatory sequence to a specific TSS, that is, TSSs are regulated by specific factors. In the former case, alternative TSSs would be expected to be always active in the same tissues and cell types, perhaps in some fixed proportion. In the latter case, alternative TSSs would be expected to show largely independent expression.

Due to the stochastic nature of gene expression, alternative TSS usage may also have functional consequences in single cells, even if they are not differentially regulated at the population level. Consider a gene with two TSSs, major and minor, which are both active. If TSSs compete for binding to regulatory elements, and activation events lead to bursts of transcription, this would lead to a stochastic, mutually exclusive, anti-correlated expression pattern, similar to

1 Stanford Cancer Institute, Stanford University School of Medicine, Stanford, CA, USA
2 Laboratory for Molecular Neurobiology, Department of Medical Biochemistry and Biophysics, Karolinska Institutet, Stockholm, Sweden
*Corresponding author. Tel: +46 8524 87577; E-mail: sten.linnarsson@ki.se

that observed for random monoallelic expression (Deng *et al*, 2014). Alternatively, if TSSs were activated independently, this would lead to uncorrelated expression in single cells. Yet another possibility is that both TSSs are simultaneously expressed, or that stochastic TSS activation occurs on a timescale much shorter than mRNA degradation, so that even in single cells these effects would be washed out, and mRNA from alternative TSSs would be correlated and detected at fixed proportions.

Heterogeneity of gene expression has been extensively studied; however, the study of single-cell isoform variation has only just started. For example, Velten *et al* (2015) recently studied single-cell polyadenylation site usage and found that even in homogenous cell populations, individual cells differ in their preferences for 3′ RNA isoform choice. Another study examined single-cell splice isoforms and found that genes having multiple splice isoforms at the population level tended to have only one expressed isoform at the single-cell level (Shalek *et al*, 2013).

Here, we address these questions using single-cell RNA-seq. We take advantage of our recently published analysis of mouse cortex and hippocampus. This extensively validated dataset comprises over 3,000 single-cell transcriptomes, classified into nine major cell types and 47 subtypes. We used STRT, a single-cell RNA-seq method suitable to study TSS usage since it captures and sequences the 5′ end of polyadenylated mRNA transcripts (Islam *et al*, 2011, 2012). The inclusion of unique molecular identifiers (UMI) ensured an increased quantitative accuracy by eliminating most PCR bias and allowed the absolute counting of mRNA molecules (Kivioja *et al*, 2012).

## Results

### Measuring TSS activity in single cells

We selected a set of 2,816 single-cell transcriptomes representing seven cell types: interneurons, somatosensory cortex pyramidal neurons, hippocampal pyramidal neurons, oligodendrocytes, astrocytes/ependymal cells, microglia, and vascular cells (endothelial cells, pericytes, and smooth muscle cells). Raw reads were mapped to the genome, assigned to FANTOM annotated TSSs, and converted to absolute number of molecules using UMIs (Islam *et al*, 2014). Henceforth, a set of reads mapped to a single genomic position, and with the same UMI, will be referred to as a "molecule" of mRNA. It should be noted that detected molecules likely represent only about 20% of all expressed molecules (Zeisel *et al*, 2015).

In order to accurately measure TSS-specific gene expression in single cells, we first needed to ensure that our protocol indeed preferentially captured 5′ ends of transcripts. We took advantage of ERCC spike-in RNAs present in each single-cell experiment. ERCC transcripts are 250–2,000 bp synthetic polyadenylated RNAs, at known concentrations. We found that 75% of all molecules mapped exactly at the 5′ end, whereas the rest were scattered across the rest of each RNA (Fig 1A).

For endogenous genes, in contrast, most molecules did not map to the annotated 5′ end. In agreement with previous findings (Islam *et al*, 2011), we found a single peak at the 5′ end and a broad 3′-biased distribution with a preference for the 3′ end (Fig 1B). As a consequence, out of the average of 26,500 detected molecules per cell, only 3,800 (14%), could be allocated to an annotated FANTOM TSS (for details, see Table EV1). Of the rest, about half mapped outside of known protein-coding genes (e.g., to expressed transposons), and the rest mapped to genes but outside the annotated TSSs (Fig 1C). The TSSs clearly showed an elevated signal, as expected. However, many molecules mapped all over the gene, including at low levels in intron regions, and some genes showed extensive 3′ UTR expression (Appendix Fig S1). Nevertheless, these findings suggest that there was enough signal specifically at the TSSs for an analysis to be possible.

To determine whether molecules assigned to TSS regions were specific, we examined a region of ± 100 bp around TSSs (here defined as the center of the FANTOM5 TSS interval). Reassuringly, we found a distinct, sharp peak at the ± 0 position, and 97% of all molecules mapped within 50 bp of the putative TSS (Fig 1D). It should be noted that FANTOM TSS intervals are not guaranteed to be centered on the true TSS, so some of the imprecision can be attributed to imperfect annotation.

Molecules mapping outside annotated TSSs could represent degradation products, as (in contrast to CAGE) our methods are not selective for capped 5′ ends. We therefore searched for signs of translation-associated mRNA decay (Pelechano *et al*, 2015), which should lead to a 3-bp repeated pattern aligned with the reading frame. We found no evidence of such degradation in this data, as can be seen in Appendix Fig S2A and B. However, a clear pattern emerged around the start and stop codons. Many molecules (i.e., 5′ ends of transcripts) mapped upstream of the start codon, but not downstream, probably reflecting the fact that most molecules in this region represent bona fide 5′ ends of transcripts (by definition, the true TSS cannot be placed after the start codon). In agreement with this hypothesis, the increase in reads upstream of the TSS closely reflected the prevalence of annotated transcription start sites. A similar, but weaker, pattern was observed around the stop codon. Interestingly, there was an enrichment of mapped reads starting just upstream of the stop codon, which may reflect pausing of the translational machinery at this point. Translation-associated mRNA decay would catch up with the stalled ribosome, leading to a relative depletion of upstream fragments and enrichment of fragments downstream of the stop codon.

To ensure that true TSS and not degradation products were assessed, molecules were counted both upstream and downstream of the TSS within a region of similar size, hereafter referred to as a "reference region". If the number of molecules in the reference region on either side of the TSS constituted more than 20% of the combined TSS and reference region reads, then the TSS was removed from the analysis. This removed around 20% of the TSSs.

Finally, to assess the reproducibility of the data, we compared correlations of gene expression profiles calculated from whole-gene bodies, as in Zeisel *et al* (2015) or only from the major TSS, both for combined and single-cell data (Fig 1F–H). As expected, data from independent pools of randomly selected cells were highly correlated, and the correlation dropped slightly for single cells. Surprisingly, for some genes, expression from the major TSS was more highly correlated compared to expression from whole-gene bodies, despite the fact that major TSSs only contained a fraction of the molecules. As shown in Appendix Figs S3A–C, the effect was rather common and depended in a large part on the expression level of the cells. For two cells with combined high average TSS expression (> 1,000

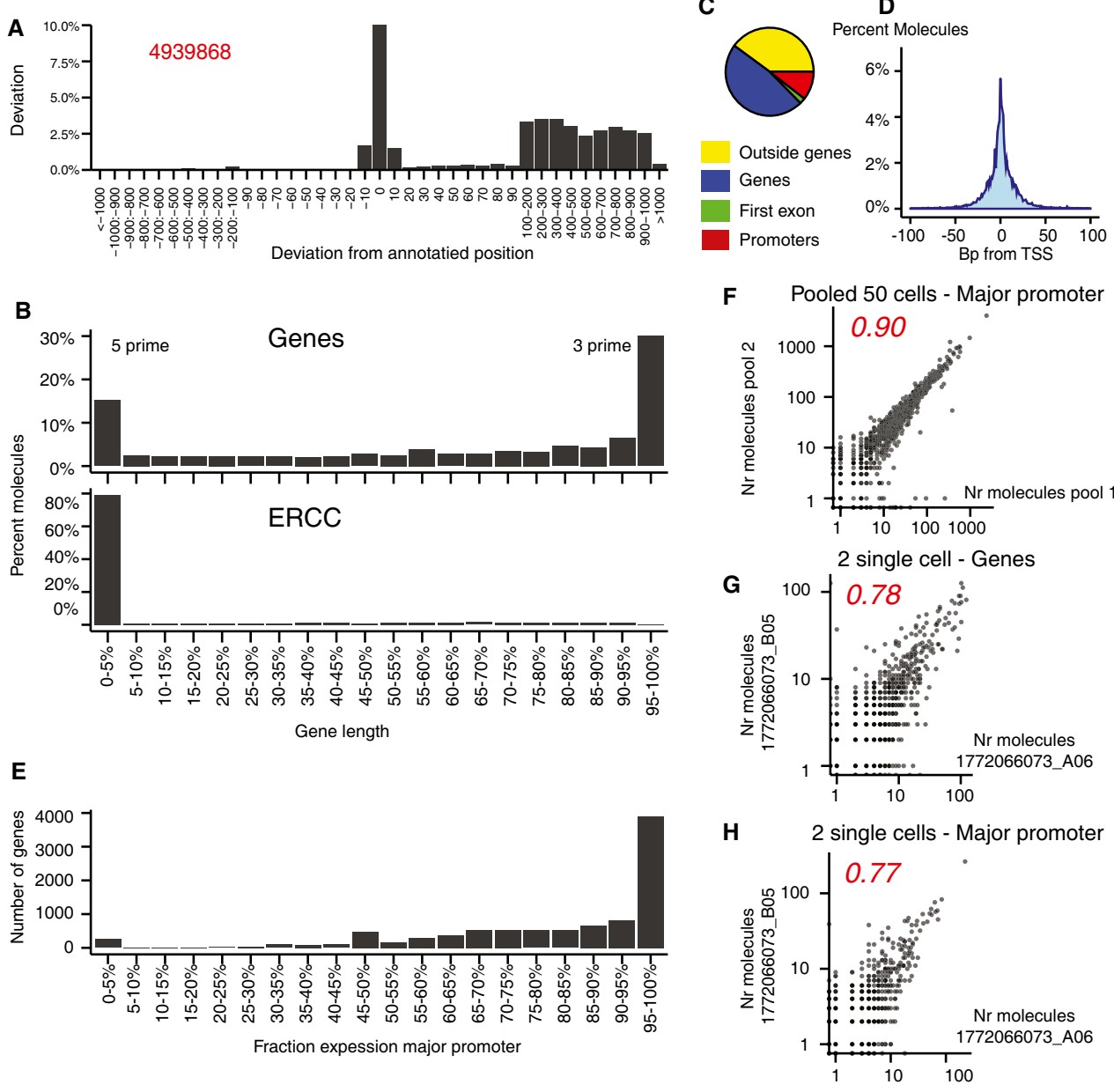

**Figure 1.    Measuring TSS activity in single cells.**

A    Number of reads deviating from starting position in ERCC control molecules shown as percent of reads mapping exactly to the starting position. The number in red shows total number of ERCC molecules participating in libraries for CA1 neurons. Note that the scale is broken.

B    Read distribution across genes, shown as percent mapped reads (after UMI correction) in 20 length-bins for CA1 neurons (top) and ERCC control RNA (bottom).

C    Read distribution for CA1 neurons. A read is only assigned once to one of the four groups.

D    CA1 neuron reads mapped to the region ± 100 bp from the TSS, defined as the center of CAGE-annotated TSSs.

E    Major TSS preference in pooled single cells from CA1 neurons.

F    Correlation of major TSS expression in pooled single CA1 neurons.

G    Correlation of whole-gene expression in individual single CA1 neurons.

H    Correlation of major TSS expression in individual single CA1 neurons.

Data information: (F–H) Pearson correlation values in red.

molecules), using major TSS counts instead of total gene counts tended to show higher correlation. This was also true when comparing two genes as shown in Appendix Figs S3D–F. The phenomenon can be explained if reads mapping to the major TSSs only reflect new transcription, while reads mapping to the total gene body reflect a number of processes that are not always correlated, such as mRNA degradation, PCR strand invasion (Tang *et al*, 2013), alternative TSS expression, intronic reads, and cryptic 3′ UTR expression.

However, we also noted a few cases where the major promoter was located outside the Refseq gene annotation leading to artefactual low correlation when using gene counts. An example of two genes that show medium correlation ($r = 0.49$) when counting major TSS reads and low correlation ($r = 0.12$) when counting total gene reads is shown in Appendix Fig S4.

## Alternative TSS usage in single cells

Having established the quality of the data, we first asked how often genes carried more than one active TSS. As shown in Fig 1E for CA1 pyramidal neurons (hereafter called CA1 neurons), around 40% of all genes showed only or almost only expression from the major TSS, counting all genes with at least one molecule mapped to a TSS. For remaining genes, there was a varying degree of co-expression between major and minor TSS. The same trend held true also for a more conservative subset of genes with expression higher than 1 molecule per cell (Appendix Fig S5). Thus, expression from two or more TSSs is common, but many genes show a strong preference for the major TSS.

Given that two TSSs are active in a cell population, it is natural to ask whether this could be explained by subpopulations of cells, or whether both TSSs are typically simultaneously active in single cells (Fig 2A). To address this, we examined the correlation of expression from the major and minor TSS between individual cells, across genes. If alternative TSSs were expressed in distinct subsets of cells (whether stochastically or by some regulated mechanism), we would expect them to be anti-correlated. In contrast, if they were simultaneously expressed in individual cells, they would be uncorrelated or positively correlated, depending on the rate of transcript degradation.

We found strong positive correlation within all studied cell types for highly expressed genes (Pearson correlation > 0.5 for most genes expressing > 4 molecules per TSS per cell in average) and a weak positive correlation for lowly expressed genes (Pearson correlation 0.1–0.5 for most genes expressing < 4 molecules per TSS per cell in average). In fact, the correlation between the TSSs increased almost linearly with expression (Fig 2E and Appendix Fig S6), indicating that at low levels of expression, noise takes over and reduces the correlation.

*Snap25*, which encodes a synaptic vesicle membrane fusion protein, was highly expressed in most cells and showed a high major/minor TSS correlation ($r = 0.80$), but also lowly expressed genes like *Dcn* were sometimes highly TSS correlated ($r = 0.85$ Fig 2B and Appendix Fig S7). Genes with very weakly correlated major/minor TSSs ($r \sim 0.1$) were the exception, for example, *Syt1*, and they did not show anti-correlation. These exceptions were probably the result of low expression. For genes with high ratio of major to minor TSS, the major TSS was consistently higher expressed in almost every cell (e.g., *Dcn*, *Snap25*, and *Son*, Appendix Fig S8).

One possible explanation of the finding that alternative TSSs are positively correlated in single cells is that the correlation would depend on degradation of the major TSS, resulting in artefactual reads on the minor TSS. In this case, a higher expressed major TSS would create more artefactual reads on the minor TSS and hence create a false correlation. An argument supporting this reasoning is that most annotated TSSs are less than 100 bp apart. Indeed, the majority of the TSSs that are expressed at the population level

represented such spatially connected TSSs. 56% of the annotated TSS in the FANTOM dataset are located in such composite transcription initiation regions (Forrest *et al*, 2014). We argue that the positive correlation is not due to degradation based on the following observations: First, TSSs with reads mapping to the reference region, constituting putative degradation events have been removed. Reads on valid TSSs were highly specific. Second, if correlation depended on degradation of reads expressed from the major TSS, then there should be an increase of genes where the major TSS was upstream of the minor TSS. However, this was not the case, and there was an almost even distribution between the major TSS being upstream or downstream of the minor (major TSS upstream of minor $n = 88$, downstream of minor $n = 109$, see Appendix Fig S9A). Third, if the correlation depended mostly on degradation of the major TSS, then the case where the major TSS is upstream of the minor should have a higher correlation. The difference was not significant ($P = 0.11$, Student's *t*-test, major TSS upstream average $r = 0.46$, SD $= 0.18$, major TSS downstream average $r = 0.42$, SD $= 0.18$, see Appendix Fig S9B). Fourth, TSSs were chosen based on CAGE peaks, an orthogonal method to single-cell RNA-seq. It is unlikely that degradation peaks by chance arises at annotated TSS regions. Fifth, genes with TSSs located far apart showed similar correlation pattern as proximal TSSs (Fig 2E) and were only slightly less correlated (Fig 2D).

Another possible explanation to the positive correlation is that the correlation is an artifact of using read counts, and would depend on sequencing depth. To verify that this is not the case, the same analysis was carried out as in Fig 2E using reads per 10 k (rpk, same as reads per million but multiplied with 10,000 instead of a million for readability) instead of counts. The number of molecules per gene per cell is strongly influenced by the total number of molecules per cell (Appendix Fig S10), which strongly influence noise (example for *Snap25*, Appendix Fig S11A–D). Therefore, only highly expressed cells (> 2,000 molecules) were included in the analysis. For most genes expressing two TSSs, the major and minor TSSs were still positively correlated after normalization (Appendix Fig S11E and F). Correlation was slightly lower after normalization, indicating that some of the correlation indeed could depend on the choice of using read counts instead of rpk, but we believe the difference in expression level between cells to be more influenced by cell size than read depth and therefore prefer to use read counts.

We next sought to quantify for each gene what percentage of cells agreed with our hypothesis that the major and minor TSSs show a fixed ratio. To this end, we used the two-sided binomial test to search for cells whose proportion of major and minor expression significantly deviated from the average proportion among all cells. For most genes, only few CA1 cells deviated from the expected ratio as can be seen in Fig 2F, and this was true for other cell types as well (Appendix Fig S12A). On average 1–3% (depending on cell type) of all cells expressing at least one molecule in any TSS per gene deviated significantly from the expected TSS ratio. However, for the gene cystatin C (*Cst3*), 44% of cells deviated from the expected ratio (Fig 2F), indicating an unusual bimodal expression pattern, and this was true for other cell types as well, albeit at varying degree (Appendix Fig S12B). Notably in vascular endothelial cells, this pattern could not be discerned, and in oligodendrocytes, proteolipid protein 1 (Plp1) showed even more cells deviating from

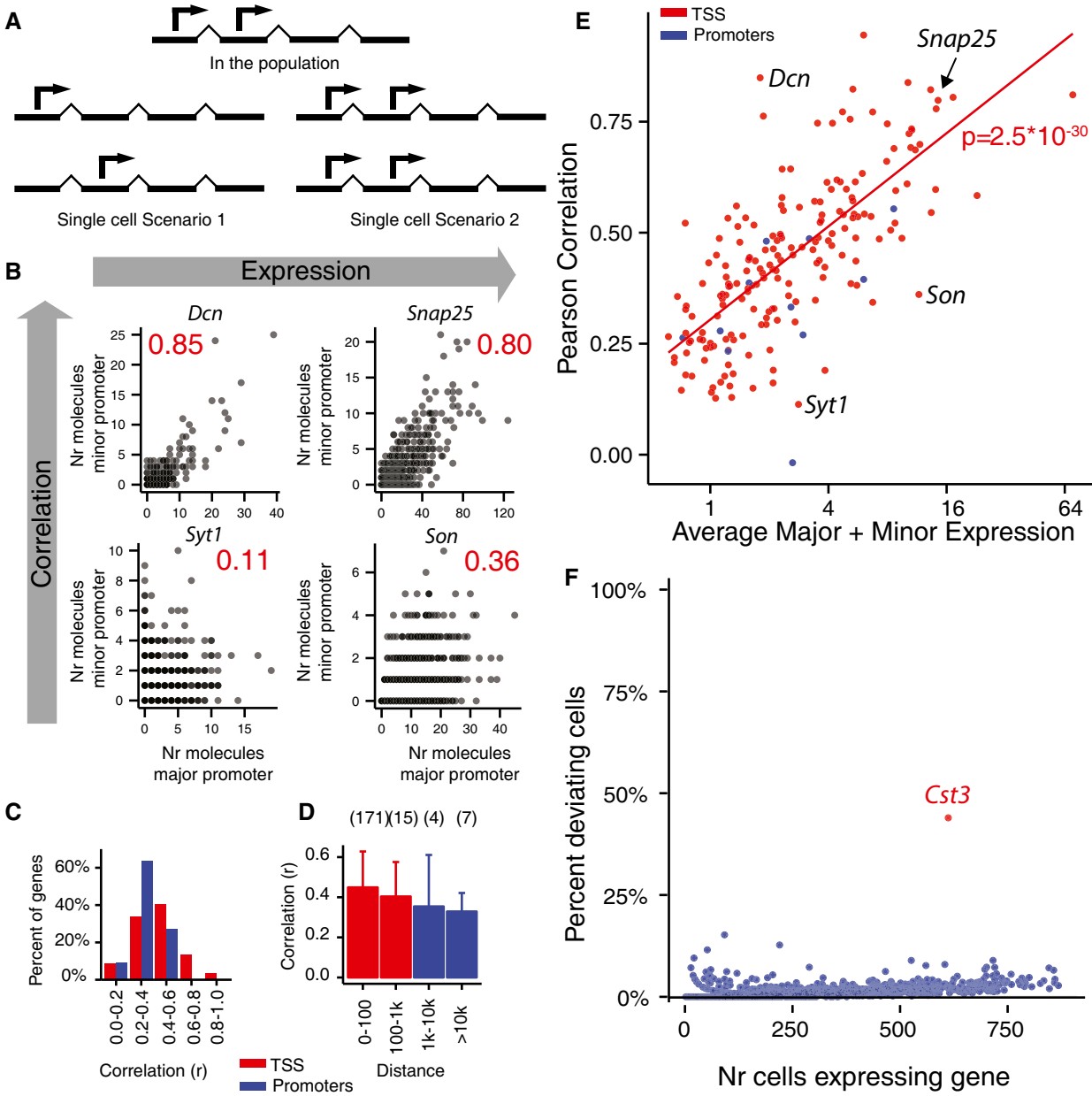

**Figure 2.   Correlated expression of major and minor TSS in single cells.**

A   Illustration of two models for TSS expression in single cells versus their common effect on the population expression.

B   Examples of TSS expression in single CA1 neuron cells. Plots show the number of mRNA molecules detected from the major and minor TSSs in single cells. Each dot is a single cell. Pearson correlation values are indicated in red. Four examples are shown, representing high and low expression, and high and low correlation.

C   Distribution of correlation values showing effect of TSS distance. Histogram based on 197 genes with an expression of at least 0.3 molecules per cell per TSS for CA1 neurons. Genes were divided based on major/minor TSS distance ("TSS", 186 genes with < 1 kb inter-TSS distance; "Promoters", 11 genes with > 1 kb TSS distance).

D   Average correlation between major/minor TSS as a function of TSS distance. Colors represent TSSs and promoters as in (C). The error bars show standard deviation and the numbers in brackets show participating genes.

E   Scatterplot showing total expression (horizontal axis) and major/minor TSS correlation coefficient (vertical). Each dot is a gene. Colors represent genes with "TSS" and "Promoter" TSSs as in (D). Four example genes from (B) are marked in the plot. Linear regression lines are shown with *P*-values of the fit.

F   Percentage of cells that significantly deviates from expected major to minor TSS ratio as a function of cell gene expression. Percentage of deviating cells were calculated using the binomial test. Each dot represents a gene.

the expected ratio than Cst3. The bimodal expression pattern of Cst3 will be discussed in more detail below.

One of the more prominent findings in this data set was the almost complete absence of genes with expression of alternative promoters. Promoter length varies from promoter to promoter; however, 1 kb upstream and a few 100 bp downstream from the TSS is a commonly used measure, for example, in Akan and Deloukas (2008). We therefore defined two TSSs located more than

1 kb apart as putative alternative promoters and examined the correlation of alternative TSS expression as a function of the distance between TSSs. We found that promoters were significantly less correlated than TSSs located in close proximity with a small margin ($P = 0.041$, Student's *t*-test; promoters $n = 11$, average $r = 0.33$, SD = 0.16; TSS $n = 186$, average $r = 0.44$ and SD = 0.18; Fig 2D and E), but the absolute difference in correlation was small. 7,369 TSSs had an expression of at least 100 molecules across all 2,816 cells. Of these, 5,872 TSSs passed the criteria that the peak at the TSS should be specific. 922 genes had expression of at least two such TSSs and are listed in Table EV2 for CA1 cells. Of these, a modest number (197) of TSS pairs expressed more than an average 0.5 molecules per cell per TSS, and only six were potential alternative promoters (*Cox16*, *Nrxn1*, *Meg3*, *Fis1*, *Grm5,* and *2610017I09Rik*). Alternative TSSs for *Cox16* and *2610017I09Rik* overlapped other genes and can therefore not with certainty be said to constitute true alternative TSSs.

All genes with potential alternative promoter usage were expressed from different exons. This was also true for two genes with alternative TSSs located within 1 kb (*Snca* and *Caly*), perhaps indicating alternative promoters. Of the coding genes, the majority (*Nrxn1*, *Grm5*, *Snca,* and *Caly*) are involved in neuronal signaling and the two other (*Cox16* and *Fis1*) are active in the mitochondria. Only *Nrxn1*, *Cox16*, and *Fis1* codes for different protein products. Four genes with TSS expressed from different exons are shown in Fig 3A–C.

The FANTOM data contain another class of CAGE peaks, located in genes and often in the 3′ UTR which are not annotated as TSSs (see Appendix Fig S1). Interestingly, we found that the correlations between alternative TSSs where one TSS was located in the 3′ UTR was much lower than when TSSs were located in the 5′ end. In these cases, correlation also did not increase with gene expression (Appendix Fig S13). This indicates that a different mechanism of regulation controls the appearance of cryptic 3′ UTR CAGE peaks. For this reason, like FANTOM, we have not included these CAGE peaks as true TSSs. Thus, our data recapitulate the surprising finding done by CAGE sequencing that low aggregates of molecules map to internal exons and sometimes rather large aggregates of molecules map to the 3′ UTR (Carninci *et al*, 2006).

An interesting observation is that genes commonly were expressed in distinct peaks from multiple genomic nucleotide positions within an annotated TSS in single cells, which was the case with, for example, *Snap25*, *Stmn3*, and *Calm1* (Appendix Figs S1 and S14). However, more often reads were scattered across the TSS region and rarely a single peak from a single nucleotide position was seen.

To verify our finding that multiple TSSs are expressed in single cells, we used a previously published dataset where six single oligodendrocyte cells were sequenced for full-length mRNA using PacBio long read technology. Due to the low sequencing depth of PacBio sequencing, not all oligodendrocyte alternative TSS could be verified. A handful of genes with alternative TSS are shown in Appendix Fig S15A–C, and for clarity, genes with long distance between the TSSs were chosen.

Considering the bursty nature of gene expression, the rather low efficiency of single-cell RNA-seq, and the low TSS mapping, we assumed that either there would be no correlation between the TSSs or that they would be anti-correlated due to the bursts. In contrast,

we found a high correlation at the single-cell level even at rather low levels of average TSS expression. In conclusion, minor TSSs were generally expressed at a fixed fraction of the major TSS, regardless of the inter-TSS distance, suggesting that they respond to common distal regulatory signals. Few genes exhibited a different expression pattern, and the most prominent of those were *Cst3*.

### Bimodal expression pattern of cystatin C (Cst3)

Cst3 encodes cystatin C (Cst C), which is a member of the cystatin superfamily and its most abundant and potent inhibitor of the cysteine cathepsins. Since Cst C can be secreted, it confers cysteine protease regulation both intra- and extracellularly. Cst C has been implicated in a number of conditions including apoptosis, antigen presentation, atherosclerosis and pathogen invasion, and the level of Cst3 mRNA can be influenced by different stimuli like inflammatory cytokines, pathogens, growth factors, hormones and oxidative stress (Xu *et al*, 2015).

For CA1 neurons, cells with more than five expressed Cst3 TSS molecules (in total 251 cells) could be clearly divided into two groups: One group had a high major TSS expression (> 2:1 ratio major/minor, here referred to as Cst3 major high) from the group with a low major TSS expression (≤ 2:1 ratio major/minor, here referred to as Cst3 major low), and this separation was consistent in other cell types as well (Appendix Fig S16A). Interestingly, by this separation, the vast majority of astrocytes were labeled as Cst3 major high, and the vast majority of interneurons were labeled Cst3 major low. Transcription factors with binding sites close to the Cst3 gene include Myog, Spi1, Ebf1, Foxo1, Stat5a:Stat5b, AR, and AP1. Of those transcription factors, only androgen receptor (AR) and activator protein 1 (AP1), which is a transcription factor that is composed of several proteins from the Jun, Fos, ATF, and JDP families, were expressed at a level higher than 0.5 molecules per cell in cells expressing Cst3, and of those, only Fos was significantly differentially expressed between Cst3 major high and Cst3 major low cells with a fold change > 2 (3.8 molecules in average for Cst3 major high compared to 1.9 for Cst3 major low, $P = 0.02$ Welch's *t*-test), which indicates that AP1, which is a transcription factor that regulates gene expression in response to stress, growth factors and cytokines, may be responsible for the high expression of Cst3 major TSS in some cells. However, the significance of the differential expression was removed after Bonferroni correction for multiple testing.

For CA1 neurons, 7,907 genes were expressed at more than 0.5 molecules per cell among Cst3 major high and major low cells. Of those, 493 genes were significantly differentially expressed in Cst3 major high ($P < 0.05$, Welch's *t*-test with Bonferroni correction) and 127 higher expressed in Cst3 major low. Gene Ontology terms associated with regulation of cell proliferation and developmental processes, receptor, and transmembrane receptor activities and with extracellular space and plasma region were significantly enriched (Table EV3) among these genes. Single-cell expression of Cst3 for CA1 neurons is shown in Appendix Fig S16B.

## Discussion

We have examined the use of alternative TSSs in single cells, using a large dataset comprising thousands of cells and found that

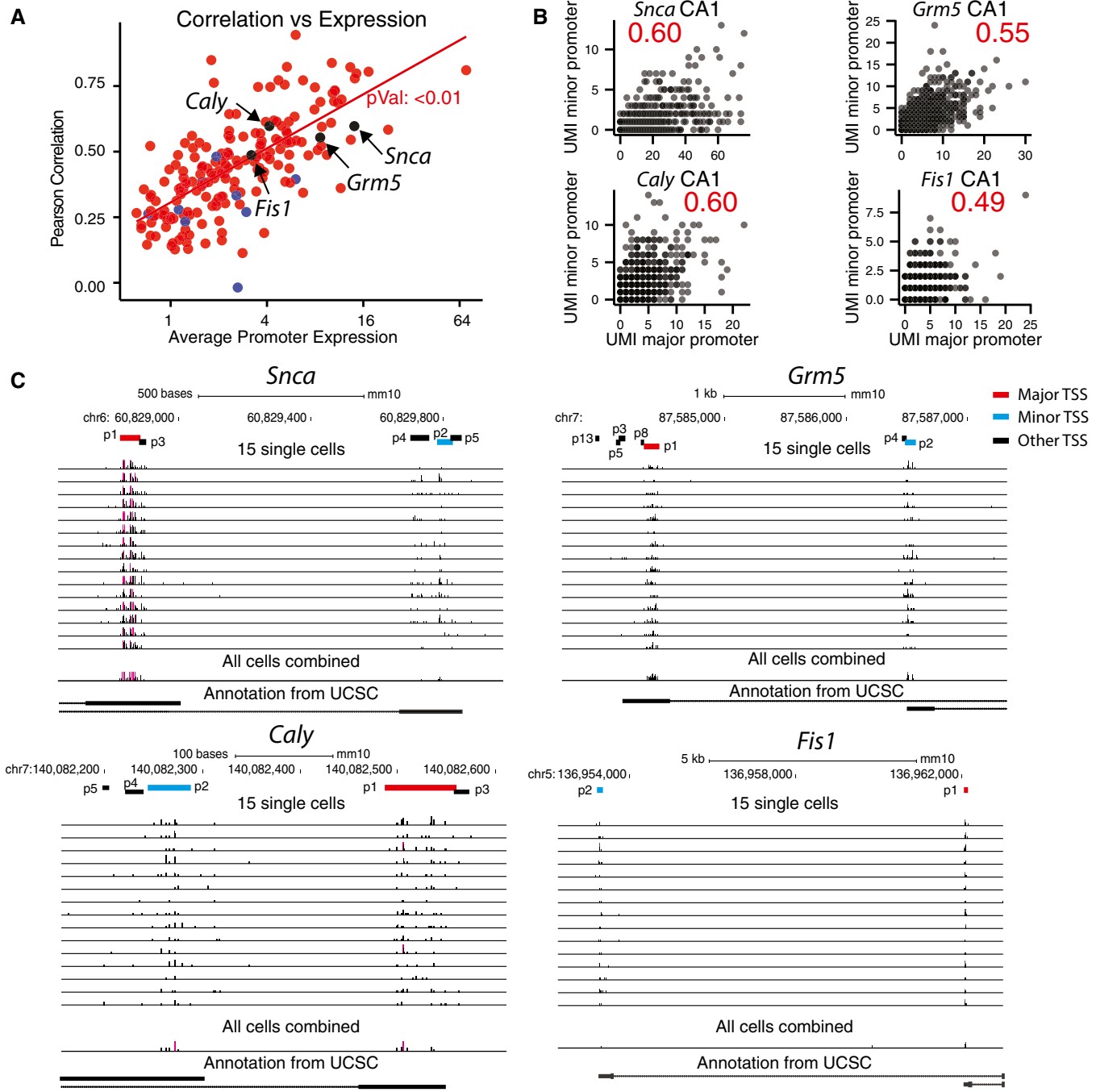

**Figure 3. TSS pairs expressed from different exons.**

A   Scatterplot showing total expression (horizontal axis) and major/minor TSS correlation coefficient (vertical). Each dot is a gene. Colors represent genes with "TSS" and "Promoter" TSSs.

B   Examples of TSS expression in single CA1 neuron cells. Plots show the number of mRNA molecules detected from the major and minor TSSs in single cells. Each dot is a single cell. Pearson correlation values are indicated in red.

C   Promoter expression of genes with TSS pairs expressed from different exons using the UCSC genome browser. Expression is shown as bars where the y-axis for single cells has a limit of five molecules and for all cells combined has a limit of 500 molecules. Major and minor TSSs are marked in red, while other CAGE peaks associated with a gene are marked in black.

alternative TSSs were almost always co-expressed in single cells. Furthermore, in highly expressed genes, alternative TSSs were expressed in a correlative manner, and the level of correlation was highly dependent on expression level indicating that the correlation in lowly expressed genes was reduced due to noise and would potentially increase with higher mRNA capture efficiency. mRNA

degradation is likely contributing, but not substantially, to the high correlation in expression between alternative TSS.

These findings would seem to contradict previous reports on the highly stochastic and bursty nature of gene expression in mammalian cells (Raj *et al*, 2006; Raj & van Oudenaarden, 2008). However, first of all, these previous studies did not measure TSS-specific gene expression, and indeed, we found that often whole-gene expression was noisier than TSS-specific expression, especially for highly expressed genes (Fig 1G and H, and Appendix Fig S3). Second, we have examined the entire transcriptome, whereas imaging-based studies have been limited to studying specific selected genes, which may or may not have been typical of the average gene. Third, it should be noted that we have here measured steady-state levels, not new transcription, and thus, the rate of mRNA degradation will influence our measurements. If the degradation rate is low, any fluctuations in transcriptional rate will tend to be erased by time averaging. Finally, there are many other possible sources of variation in observed total gene expression, such as the total number of mRNA molecules per cell (which varies substantially), and technical differences such as differences in sequencing depth or mRNA recovery. Indeed, the expression level of individual genes is dependent on total mRNA expression level in a cell (Appendix Fig S10). These other factors would not be expected to differ for alternative TSSs of the same gene.

There are at least two possible explanations for the high correlation between alternative TSSs: Either gene expression is not as bursty as previously believed (relative to the degradation rate), or both TSSs participate in each burst of transcription. Our finding that TSSs located far apart were less correlated lends some support to the latter explanation, although the effect was small.

Very few genes were exceptions to the rule that there is a set ratio between the expression of major and minor TSS. The most prominent of these genes was *Cst3* which codes for the gene cystatin C, an inhibitor of cysteine proteases. mRNA of *Cst3* showed a bimodal expression pattern where the major and minor TSS were expressed to a similar degree in some cells, while in other cells, the major TSS was highly selectively expressed. Genes that were highly expressed in the latter cells were associated with GO terms for receptor activity, extracellular space, and the plasma membrane, possibly indicating response to a stress signal. The immediate-early transcription factor Fos was associated with selective major TSS expression of *Cst3*, is a well-known stress response factor, and is regulated by neuronal activity in the brain.

For highly expressed genes, the correlation between two different genes was higher when using only reads mapping to the major TSS as compared to using reads mapping to the full gene body (Appendix Fig S3). This is probably due to the fact that reads mapping to the major TSS only reflect gene expression, while reads mapping to the gene reflect many processes that may not always be correlated, including mRNA degradation, alternative TSS expression, PCR strand invasion, intronic reads, and cryptic 3′ expression. This may have implications for clustering of single cells into cell types since many clustering methods rely on correlation between genes.

Surprisingly, few genes expressed alternative promoters (defined as TSS located more than 1 kb apart), and only three coded for different protein products. One explanation for this may be that there are few occasions where it would be beneficial for a gene to express transcripts from multiple promoters within a cell type. The presence

and levels of TFs vary across tissues and developmental time, and since it is known that specific TFs associate more strongly with certain promoters it is reasonable to believe that this can influence the promoter preference for specific genes (Rach *et al*, 2009). Similarly alternative promoters are known to be expressed across tissues and developmental stages and can, for example, ensure that housekeeping genes keep a similar expression level given a different regulatory landscape, or they can tune the level of expression between different cell types (Ayoubi, 2005). However, the need for a single cell type to express multiple TSS isoforms of a gene may be limited.

In summary, we found a surprising degree of co-expression of alternative TSSs in single cells. These findings provide strong constraints on models of transcriptional regulation.

# Materials and Methods

### Data collection

This study uses previously published data (Zeisel *et al*, 2015, Gene Expression Omnibus www.ncbi.nlm.nih.gov/geo under accession code GSE60361) from 2,816 single cells from the mouse somatosensory cortex and hippocampal CA1 region of genetically outbred (CD-1) mice. Raw reads were remapped to mm10 using Bowtie I, allowing for three mismatches, and annotated to TSSs as explained below. Reads were converted into mRNA molecule counts using UMIs, as previously explained (Kivioja *et al*, 2012). The UMI sequence is 6 bp long, and reads with the same UMI sequence were collapsed and UMIs with only one read were removed. Number of molecules per cell can be found in Table EV4.

Full-length single oligodendrocyte mRNA sequencing data were taken from previously published data (Karlsson & Linnarsson, 2017, Gene Expression Omnibus www.ncbi.nlm.nih.gov/geo under accession code GSE76026).

This link: http://genome-euro.ucsc.edu/cgi-bin/hgTracks?hgS_do OtherUser = submit&hgS_otherUserName = Kasper&hgS_otherUser SessionName = mm10_public_promoters_CA1neurons provides access to a UCSC track showing promoter expression of 20 single cells as well as the combined expression for all CA1 neurons.

This link: http://genome-euro.ucsc.edu/cgi-bin/hgTracks?hgS_do OtherUser = submit&hgS_otherUserName = Kasper&hgS_otherUser SessionName = mm10_public_promoters_CA1_Oligos provides access to a UCSC track showing promoter expression of 10 single CA1 neuron cells, 10 single oligodendrocyte cells as well as the combined expression for all CA1 neurons and all oligodendrocytes.

### Analysis

#### Input data

Transcription start site regions were defined based on an early access program from the FANTOM 5 project and may therefore differ slightly from the published FANTOM 5 mouse TSS database (Forrest *et al*, 2014). All CAGE tags used are shown in Table EV5. CAGE tags were curated, and tags without association to a gene were discarded. CAGE tags were also moved from mm9 to mm10 using liftOver from UCSC. In the cases where two TSS regions overlapped, the longer TSS region was shortened so there would be no overlap and the minimum distance between two TSSs was 1 bp.

The distance between TSSs was calculated as the distance between the edges of the TSSs, not the distance between the centers of the TSSs. Molecules were allocated to TSSs using a custom program. For each single cell, cDNA molecules with their 5′ end within the curated CAGE-defined FANTOM 5 TSSs were counted.

For each gene, the major TSS was defined as the TSS that had the greatest number of mapped mRNA molecules (UMIs) across the entire dataset (i.e., counting all cell types). The minor TSS was defined as the TSS that had the second-largest number of molecules. Because of this definition, the major TSS need not be the highest expressed TSS in individual cells or cell types.

The major and minor TSSs of each gene were considered to be proximal TSS if the distance between the TSSs was < 1 kb; all other TSS pairs were considered as promoter pairs.

To limit the interference of degradation on the analysis, mapped reads were calculated in a region of equal size as the TSS next to the annotated TSS both upstream and downstream (here called a reference region). If another TSS was located in the area where the reference region should be, then the reference region was moved outside of that TSS. If a reference region (either upstream or downstream) contained more than 20% of the combined reads from the TSS and reference region, then that particular TSS was removed from the analysis.

Normalization to rpk was done for each cell as follows: molecules per TSS/total expressed TSS molecules × 10,000.

RefSeq genes used for annotation in Fig 1B and G were downloaded from the UCSC table browser (21-09-2015), and only the first isoform by the order of the file of each gene was kept.

We used a slightly modified database to map ERCC reads, and the modifications are shown in Table EV6. The reason for this is that we previously have observed that for many ERCC reads, the 5′ end is slightly upstream of the ERCC reference sequences. In Fig 1A, for each ERCC, the median of all mapped reads was used as the starting position and the percentage of reads deviating from the starting position is shown.

Transcription factor binding sites were extracted from the UCSC genome browser based on the ORegAnno database, as well as from a previously published paper (Huh *et al*, 1995).

### Statistical methods

To calculate the difference in correlation between genes with alternative TSS and genes with alternative promoters, the non-parametrical a two-sided Mann–Whitney *U*-test was used, implemented by the R function wilcox.test since the number of genes with alternative promoters was too few to ensure normal distribution ($n = 11$), but had a similar variance. To calculate the difference in correlation between the upstream and downstream location of the major TSSs (compare with Appendix Fig S7), a two-sided Student's *t*-test was used, implemented with the R function t.test, since the data were normally distributed. The line and associated *P*-value for Fig 2E and Appendix Figs S6, S7, and S9 was calculated with the lm method in R. Correlations were calculated using Pearson correlation and implemented in python using the stats.linregress function in scipy or the cor function in R.

To calculate how many cells that significantly deviated from expected major to minor TSS ratio, the binomial test was performed for each gene across all cells and implemented with the python function scipy.stats.binom_test, using number of reads of the major TSS, the total number of TSS reads, and the percentage of major

TSS reads across all cells in a cell type as the probability of success and without correction for multiple testing.

To calculate significant alternative TSS usage between cell types, the minor fraction (molecules on minor TSS divided by the sum of molecules on both TSSs) for each gene and cell was first calculated. Cells with no expression in either major or minor TSS were removed and genes expressed in fewer than 20% of cells in any of the two cell types were removed to keep only genes with at least modest expression in the two cell types. Since the data were not normally distributed, a two-sided non-parametrical Mann–Whitney test was applied, implemented in python using the function stats.mannwhitneyu in scipy, with Bonferroni correction on the minor fraction for each gene on the two cell types. The data could not assume equal variance of cell expression between cell types for each gene; however, the combined sample sizes were large ($n \geq 75$) for all genes with a statistical significant differential usage of minor fraction. The number of cells was more than 800 for some genes and cell types, so small differences in minor fraction could be determined as statistically significant. Therefore, we additionally required a difference of at least 0.3 in minor fraction to find genes with potential biological significance.

All examples from a single cell type were taken from CA1 neurons since they had a high number of molecules per cell in average and constituted a large fraction of the total dataset with 876 single cells. Comparisons between cell types were performed between CA1 neurons and oligodendrocytes, unless otherwise stated, because there were many oligodendrocyte cells, and they are a clearly distinct cell type from neurons.

**Expanded View** for this article is available online.

### Acknowledgements

We acknowledge funding from the Swedish Research Council (STARGET) and the European Research Council (ERC grant 261063, BRAINCELL).

### Author contributions

SL conceived the study, PL mapped sequencing reads to the genome, and KK performed all additional analysis. KK and SL drafted the manuscript with input from PL.

### Conflict of interest

The authors declare that they have no conflict of interest.

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
