## [Review Process File · Molecular Systems Biology]

Alternative TSSs are co-regulated in single cells in the mouse brain

Kasper Karlsson, Peter Lönnerberg, Sten Linnarsson

Corresponding author: Sten Linnarsson, Karolinska Institutet

Review timeline:

Submission date:	07 October 2016
Editorial Decision:	02 December 2016
Revision received:	02 March 2017
Editorial Decision:	03 April 2017
Accepted:	11 April 2017

Editor: Maria Polychronidou

Transaction Report:

1st Editorial Decision

02 December 2016

Thank you again for submitting your work to Molecular Systems Biology. We have heard back from two of the three referees who agreed to evaluate your manuscript. As you will see below, the reviewers raise substantial concerns on your work, which unfortunately preclude the publication of the study in its current form.

In particular, the reviewers mention that additional analyses are required i) to better support the main findings and ii) to provide conclusive novel insights. However, considering that they appreciate that the addressed topic is timely and important for the field, we would like to offer you a chance to revise and extend the study and address the points raised.

Without repeating all the points listed below, the most fundamental issues that need to be addressed are the following:

- Additional analyses should be performed to better support the existence of multiple TSSs. Both reviewers provide constructive suggestions in this regard.

- Follow up analyses on the reported findings (i.e. Cst3 alternative TSS preference in the different cell types) would significantly enhance the impact of the study. Reviewer #2 provides suggestions and lists (see point #2) other findings that could alternatively be followed up in order to provide novel insights.

REFEREE REPORTS

Reviewer #1:

The manuscript entitled "Alternative TSSs are co-regulated in single cells in the mouse brain" is a follow up analysis of previously published Zeisel A., et al "Cell types in the mouse cortex and hippocampus revealed by single-cell RNA-seq". Understanding of TSS usage in single cells and also across different cell types is essential step to dissect the regulatory process surrounding promoters and this manuscript uses the single cell sequencing data to address this important question. The authors report that alternative TSS usage is a regulated process and the correlation between two TSS expressed in single cells of the same cell type is surprisingly high. The authors explore potential scenarios where TSSs observed in single cells are not derived from artifacts but potentially from coordinated bursting. However, additional supporting evidence to demonstrate the co-existence as well as regulatory connections of alternative TSSs in single cells would greatly strengthen the claims made in this manuscript.

Comments

Alternative methods to validate multiple TSSs based on 5'-RACE (e.g. single cell PCR based) or single-molecule FISH of alternative TSSs would strongly substantiate the claim that multiple TSSs co-exists in a single cell. For instance, different fluorescently labeled probes targeting alternative TSSs with larger spans (e.g. ~5 kb) would clearly demonstrate that two transcripts with different TSS co-exist in a single cell.

It would be imperative that the observations made in this study is not affected by potential doublets innate to the microfluidics system used in the original Zeisel a., et al. Science 2015 paper. Is the co-existence of alternative TSSs greater than the expected number of doublets found in the study and can the authors demonstrate statistical significance of its enrichment to single (stand alone) TSSs?

Distinct binding sites of transcription factors often control alternative TSSs. Do authors identify putative transcription factor binding sites (TFBS) in the proximity of alternative TSS that would suggest any possible explanation for its co-activation?

As related to the previous point, figure 4A illustrates that the significant alternative TSS usage is contextual to the cell type. This seems to suggest that differential regulatory elements (TFBS, transcription factors) are regulating cell-specific TSS expression. Do authors find additional evidence of regulatory components in these alternative TSSs (in the same single cells and across different cell types)?

Is there any functional distinction between alternative TSS vs alternative promoters defined in this study?

Reviewer #2:

The authors leverage their recent 5' single cell data of single mouse cortex and hippocampal cells to examine regulation of 5' TSS usage at single cell resolution. They find that the ratio of alternative TSS usage is largely conserved across single cells, and (with a small number of exceptions), found little evidence for stochastic or mutually exclusive TSS selection.

The paper addresses an important and previously unanswered question - the coordination of 5' TSS across single cells within the same type, as well as across cell types. Indeed, the dataset leveraged here is unique in its capability to answer this question, as the STRT technique captures 5' TSS sites, and the use of UMIs enables the author to control for extensive PCR jackpotting that can be highly confounding. However, I was unable to discern a clear message from the manuscript, and have significant concerns about the analytical framework used to assess TSS regulation.

Main comments

1. Overall, I thought the authors did a good job with annotating single cell reads to individual TSS.

This type of analysis has many caveats, but the QC and description shows that this was done carefully and accurately.

2. I did struggle to come away with a clear set of messages from the manuscript. Many of the results are presented anecdotally (i.e.: Interestingly,.....which may reflect...) and are not followed up on in detail. Examples include:

- The potential for 5' TSS reads to more accurately reflect new transcription
- Enrichment of reads prior to the stop codon, reflecting stalled ribosome dynamics in RNA-seq (?)
- Changes in correlation strength depending on the location of the TSS
- The Cst3 result (more below)

Any one of these anecdotes are potentially interesting with further follow-up analysis, although some seem quite unlikely. As is, they detract from the main story, and make it difficult to focus on the main findings.

It appears that there is a single main finding, that most single cells express multiple TSS and at similar ratios. Its not clear to me that this is particularly unexpected or exciting, but it does contribute to an open question, particularly with recent reports of greater stochasticity in splicing and 3' processing.

3. I disagree strongly with the use of correlation as a metric for measuring TSS usage. The strength of the correlation is interpreted as a measure of the regulation strength of the TSS, but I don't believe that this is a correct interpretation. Instead, the correlation is driven by different overall expression levels of the gene across single cells, perhaps due to differences in cell size or capture efficiency.

For example - suppose as a thought experiment that the authors sequenced 100 cells of identical size. Lets say for gene X total counts were ~50 for each cell. Now suppose that there are two TSS which are nearly 50/50 across each single cells (so each cell detects ~ 28-32 counts of TSS1 and TSS2). If the authors do a correlation plot of TSS1 vs TSS2 here, they will see no correlation, despite, the fact that the TSS choice is incredibly tightly regulated. Therefore, I feel that the correlation-based analyses are inappropriate to use here.

Instead, the authors should focus on the ratio of TSS usage. They use this analysis sparingly for single cells, particularly later in the manuscript, but the correlation analyses are more prominent. The challenge of using ratios, of course, is that they are difficult for cells with low molecular counts for gene X.

I want to suggest a different analysis that the authors could easily employ, that would significantly improve the manuscript. The authors suggest that a null hypothesis - where each cell expresses the TSS at the same ratio - holds for most genes and most cells. Why not test this hypothesis directly with a binomial (or multinomial) test? The authors could define a TSS ratio (based on pooled data), and then search for cells whose read data deviated from this average ratio. In this case, cells with low counts would not have significant evidence to deviate from the null model, but cells with high molecule counts could significantly deviate if they had skewed read counts. Then, the authors could explicitly test the hypothesis they state, rather than examining it in an indirect and potentially misleading way through correlation analysis.

My intuition is that the conclusions will not change significantly, but I feel this is an important analysis.

3. The Cst3 result is quite interesting, observing a subpopulation of cells that predominantly choose a single TSS. However, the section feels perfunctory, with only two 'variable TSS' genes highlighted, and is missing needed follow-up analyses.

- Are there other genes like CST3 (the binomial test discussed above could easily identify these)
- For the subpopulation that primarily expresses one TSS, is there anything else that is different (either for expression or TSS usage) across single cells between cells?
- is this relationship true across all cell types, or just the two discussed?

4. The comparison of population pools did not significantly add to the manuscript. The main finding seems to be that TSS are used similarly for similar cell types, but there are differences when

considering different tissue cell types (i.e. neuronal and immune cell types). This does not seem surprising or unexpected, and could easily have been observed with bulk data (even with low-medium frequency sorting errors)

1st Revision - authors' response

02 March 2017

Reviewer #1:

The manuscript entitled "Alternative TSSs are co-regulated in single cells in the mouse brain" is a follow up analysis of previously published Zeisel A., et al "Cell types in the mouse cortex and hippocampus revealed by single-cell RNA-seq". Understanding of TSS usage in single cells and also across different cell types is essential step to dissect the regulatory process surrounding promoters and this manuscript uses the single cell sequencing data to address this important question. The authors report that alternative TSS usage is a regulated process and the correlation between two TSS expressed in single cells of the same cell type is surprisingly high. The authors explore potential scenarios where TSSs observed in single cells are not derived from artifacts but potentially from coordinated bursting. However, additional supporting evidence to demonstrate the co-existence as well as regulatory connections of alternative TSSs in single cells would greatly strengthen the claims made in this manuscript.

Comments:

1.

Alternative methods to validate multiple TSSs based on 5'-RACE (e.g. single cell PCR based) or single-molecule FISH of alternative TSSs would strongly substantiate the claim that multiple TSSs co-exists in a single cell. For instance, different fluorescently labeled probes targeting alternative TSSs with larger spans (e.g. ~5 kb) would clearly demonstrate that two transcripts with different TSS co-exist in a single cell.

Response:

We agree that independent validation would strengthen the findings. Single-molecule FISH would be very challenging since in most cases, alternative TSSs create nested transcripts where the shorter transcript is entirely included in the longer transcript. This makes it difficult to directly detect the shorter transcript by smFISH, independent of the longer transcript (i.e. there is no unique sequence to target). 5'-RACE is essentially equivalent to the single-cell mRNA seq method we used (indeed, Clontech kits for 5'-RACE are based on exactly the same template switching technology). Thus it would not serve as a truly independent validation.

Instead, we opted to perform PacBio full-length sequencing of six single cells. PacBio sequencing confirmed alternative TSS usage in single cells for multiple genes and these findings are now included in supplementary figure 15A-C.

2. It would be imperative that the observations made in this study is not affected by potential doublets innate to the microfluidics system used in the original Zeisel a., et al. Science 2015 paper. Is the co-existence of alternative TSSs greater than the expected number of doublets found in the study and can the authors demonstrate statistical significance of its enrichment to single (stand alone) TSSs?

Answer:

This is an important point, and we have now directly tested the hypothesis that observed alternative TSSs were artefacts due to doublets. We reasoned that if alternative TSSs were predominantly observed in doublets, then genes with cell type-specific expression should show much lower rates of alternative TSS usage than genes with broader expression (given that cell type-specific genes would not be affected by doublets). However, looking only at neuron specific genes (based on a rank-sum test between CA1-neurons and oligodendrocytes), with a total gene expression greater than 1 molecule per cell in CA1 neurons, 1.93% of these genes contained an alternative TSS in CA1 neurons (where both TSS expressed at least 0.5 molecules per cell in CA1 neurons), as compared with all genes, where 1.90% of all genes contained

aTSS. Thus, we see little evidence that doublets creates a large number of false alternative TSS.

Furthermore, all cells included in this study (and our previous work) have been individually imaged at high magnification, and visible doublets have been removed. We estimate that this procedure leads to a residual rate of doublets less than 10%.

3. Distinct binding sites of transcription factors often control alternative TSSs. Do authors identify putative transcription factor binding sites (TFBS) in the proximity of alternative TSS that would suggest any possible explanation for its co-activation?

Answer:

This is an interesting suggestion, and indeed stable differences in the activity of adjacent TSSs would very likely require differences in the local regulatory machinery. However, the vast majority of TSS were located too close to each other to be able to discern if they are regulated by the same or different TF. We have now examined this possibility for the small number of expressed TSSs located far apart (>1 kb) and that showed alternative regulation, but the result was inconclusive. We note that the main finding of the paper is that simultaneously expressed, alternative TSSs, do not show independent regulation and instead maintain fixed ratios. In such cases we would not expect to find distinct TFs regulating each TSS.

However, we have expanded our analysis of the strikingly bimodal regulation of *Cst3*, and found a significant correlation with a number of genes, including the immediate-early transcription factor *Fos*, which also has a target site near the *Cst3* promoter.

4. As related to the previous point, figure 4A illustrates that the significant alternative TSS usage is contextual to the cell type. This seems to suggest that differential regulatory elements (TFBS, transcription factors) are regulating cell-specific TSS expression. Do authors find additional evidence of regulatory components in these alternative TSSs (in the same single cells and across different cell types)?

Answer:

Fig 4A shows that only a small fraction of genes carry pairs of TSSs that are expressed at different ratios in distinct cell types. Unfortunately, the short distance between TSSs and the low number of positive cases precluded an analysis of local regulatory elements.

As suggested by another referee, we have removed this section, which contains results that could have also been obtained by bulk RNA-seq.

5. Is there any functional distinction between alternative TSS vs alternative promoters defined in this study?

Answer:

We did not find any sharp distinction between two classes of alternative TSSs, but a gradual shift as a function of the inter-TSS distance (e.g. Fig 1d). We therefore used an operational distinction between alternative TSS and alternative promoters, where if the distance was <1kb they were defined as TSS and if the distance was >1kb they were defined as promoters.

Reviewer #2:

The authors leverage their recent 5' single cell data of single mouse cortex and hippocampal cells to examine regulation of 5' TSS usage at single cell resolution. They find that the ratio of alternative TSS usage is largely conserved across single cells, and (with a small number of exceptions), found little evidence for stochastic or mutually exclusive TSS selection.

The paper addresses an important and previously unanswered question - the coordination of 5' TSS across single cells within the same type, as well as across cell types. Indeed, the dataset leveraged here is unique in its capability to answer this question, as the STRT technique captures 5' TSS sites, and the use of UMIs enables the author to control for extensive PCR jackpotting that can be highly

confounding. However, I was unable to discern a clear message from the manuscript, and have significant concerns about the analytical framework used to assess TSS regulation.

Main comments

1. Overall, I thought the authors did a good job with annotating single cell reads to individual TSS. This type of analysis has many caveats, but the QC and description shows that this was done carefully and accurately.

2. I did struggle to come away with a clear set of messages from the manuscript. Many of the results are presented anecdotally (i.e. : Interestingly,.....which may reflect...) and are not followed up on in detail. Examples include:

- The potential for 5' TSS reads to more accurately reflect new transcription
- Enrichment of reads prior to the stop codon, reflecting stalled ribosome dynamics in RNA-seq (?)
- Changes in correlation strength depending on the location of the TSS
- The Cst3 result (more below)

Any one of these anecdotes are potentially interesting with further follow-up analysis, although some seem quite unlikely. As is, they detract from the main story, and make it difficult to focus on the main findings.

Answer:

We have revised the manuscript to more clearly focus on a few key findings, especially Cst3, and hope that the discussion is now easier to follow.

It appears that there is a single main finding, that most single cells express multiple TSS and at similar ratios. Its not clear to me that this is particularly unexpected or exciting, but it does contribute to an open question, particularly with recent reports of greater stochasticity in splicing and 3' processing.

Answer:

Considering the recent discovery of transcriptional bursting and as reviewer #2 mention the great stochasticity in splicing and 3' processing we think the finding of a fixed ratio between the major and minor TSS to be both rather surprising and of considerable value.

First of all, it informs current work on the 3D architecture of chromatin, and shows that there may be no need to assume that distal enhancers are highly selective for specific individual TSSs or promoters.

Second, it shows that whatever mechanism is responsible for stochastic gene expression cannot be intrinsic to individual TSSs, but must operate at the level of a set of TSSs (at minimum, pairs of TSSs, which is what we studied here).

3. I disagree strongly with the use of correlation as a metric for measuring TSS usage. The strength of the correlation is interpreted as a measure of the regulation strength of the TSS, but I don't believe that this is a correct interpretation. Instead, the correlation is driven by different overall expression levels of the gene across single cells, perhaps due to differences in cell size or capture efficiency.

For example - suppose as a thought experiment that the authors sequenced 100 cells of identical size . Lets say for gene X total counts were ~50 for each cell. Now suppose that there are two TSS which are nearly 50/50 across each single cells (so each cell detects ~ 28-32 counts of TSS1 and TSS2). If the authors do a correlation plot of TSS1 vs TSS2 here, they will see no correlation, despite, the fact that the TSS choice is incredibly tightly regulated. Therefore, I feel that the correlation-based analyses are inappropriate to use here.

Instead, the authors should focus on the ratio of TSS usage. They use this analysis sparingly for single cells, particularly later in the manuscript, but the correlation analyses are more prominent. The challenge of using ratios, of course, is that they are difficult for cells with low molecular counts for gene X.

I want to suggest a different analysis that the authors could easily employ, that would significantly improve the manuscript. The authors suggest that a null hypothesis - where each cell expresses the TSS at the same ratio - holds for most genes and most cells. Why not test this hypothesis directly with a binomial (or multinomial) test? The authors could define a TSS ratio (based on pooled data), and then search for cells whose read data deviated from this average ratio. In this case, cells with low counts would not have significant evidence to deviate from the null model, but cells with high molecule counts could significantly deviate if they had skewed read counts. Then, the authors could explicitly test the hypothesis they state, rather than examining it in an indirect and potentially misleading way through correlation analysis.

My intuition is that the conclusions will not change significantly, but I feel this is an important analysis.

Answer:

We thank you for this valuable suggestion. This analysis has now been carried out and your intuition was correct, the conclusion didn't change but was rather confirmed. Cst3 still stands out as a rare exception to the hypothesis that the ratio between the major and minor TSS is constant. A figure has been added to the main text (figure 2F) as well as supplemental figure 12. The text describing the binomial test is on page 8, paragraph 2 and well as in the methods section.

3. The Cst3 result is quite interesting, observing a subpopulation of cells that predominantly choose a single TSS. However, the section feels perfunctory, with only two 'variable TSS' genes highlighted, and is missing needed follow-up analyses.

- Are there other genes like CST3 (the binomial test discussed above could easily identify these)
- For the subpopulation that primarily expresses one TSS, is there anything else that is different (either for expression or TSS usage) across single cells between cells?
- is this relationship true across all cell types, or just the two discussed?

Answer:

Further investigations into the bimodal expression of Cst3 have been carried out. For most cell types Cst3 was the only gene with bimodal expression, however for Oligodendrocytes there are a few more genes with a prominent number of cells deviating from the expected major to minor TSS ratio, most notably Plp1 (see supporting figure 12)

Genes with differential expression between cells with high and low Cst3 major TSS expression have been identified as well as GO terms associated with those genes (supplemental table 3). It seems that cells with high Cst3 major TSS expression are more involved in inter-cellular communication. Significant GO terms include receptor activity, extracellular space and plasma membrane. A supporting table with GO terms defining high and low Cst3 major TSS expression for the four cell types with bimodal Cst3 TSS expression have been added (supporting table 3)

4. The comparison of population pools did not significantly add to the manuscript. The main finding seems to be that TSS are used similarly for similar cell types, but there are differences when considering different tissue cell types (i.e. neuronal and immune cell types). This does not seem surprising or unexpected, and could easily have been observed with bulk data (even with low-medium frequency sorting errors)

Answer:

We agree that these results could have in principle been obtained from bulk data. We have removed this section, making room for the more extensive analysis of Cst3 on page 9-10.

2nd Editorial Decision

03 April 2017

Thank you again for sending us your revised manuscript. We have now heard back from the two referees who were asked to evaluate the study. As you will see below, both referees think that the previously raised issues have been satisfactorily addressed and the study is now suitable for publication.

Before we formally accept the manuscript we would ask you to address some remaining editorial issues listed below.

REFEREE REPORTS

Reviewer #1:

The description of alternative TSS in single cells is well presented and opens new avenues to explore gene regulation at the promoter level. The authors have adequately addressed comments raised by this reviewer.

Reviewer #2:

The authors have addressed my comments with strong revisions that have substantially improved the manuscript.

Corresponding Author Name:
Journal Submitted to:
Manuscript Number: